# A Novel System for the Detection of Spontaneous Abortion-Causing Aneuploidy and Its Erroneous Chromosome Origins through the Combination of Low-Pass Copy Number Variation Sequencing and NGS-Based STR Tests

**DOI:** 10.3390/jcm12051809

**Published:** 2023-02-23

**Authors:** Caixia Lei, Kai Liao, Yuwei Zhao, Zhoukai Long, Saijuan Zhu, Junping Wu, Min Xiao, Jing Zhou, Shuo Zhang, Lianbin Li, Yijian Zhu, Daru Lu, Jingmin Yang, Xiaoxi Sun

**Affiliations:** 1Shanghai Ji Ai Genetics & IVF Institute, Obstetrics and Gynecology Hospital, Fudan University, Shanghai 200011, China; 2State Key Laboratory of Genetic Engineering, MOE Engineering Research Center of Gene Technology, School of Life Sciences, Fudan University, Shanghai 200438, China; 3Shanghai WeHealth BioMedical Technology Co., Ltd., Shanghai 201318, China; 4NHC Key Laboratory of Birth Defects and Reproductive Health, Chongqing Key Laboratory of Birth Defects and Reproductive Health, Chongqing Population and Family Planning, Science and Technology Research Institute, Chongqing 404100, China; 5Key Laboratory of Female Reproductive Endocrine Related Diseases, Obstetrics and Gynecology Hospital, Fudan University, Shanghai 200011, China

**Keywords:** low-pass copy number variation sequencing, NGS-based STR test, miscarriage samples, aneuploidy, chromosome error origins

## Abstract

During the period of 2018–2020, we first combined reported low-pass whole genome sequencing and NGS-based STR tests for miscarriage samples analysis. Compared with G-banding karyotyping, the system increased the detection rate of chromosomal abnormalities in miscarriage samples to 56.4% in 500 unexplained recurrent spontaneous abortions. In this study, a total of 386 STR loci were developed on twenty-two autosomes and two sex chromosomes (X and Y chromosomes), which can help to distinguish triploidy, uniparental diploidy and maternal cell contamination and can trace the parental origin of erroneous chromosomes. It is not possible to accomplish this with existing methods of detection in miscarriage samples. Among the tested aneuploid errors, the most frequently detected error was trisomy (33.4% in total and 59.9% in the error chromosome group). In the trisomy samples, 94.7% extra chromosomes were of maternal origin and 5.31% were of paternal origin. This novel system improves the genetic analysis method of miscarriage samples and provides more reference information for clinical pregnancy guidance.

## 1. Introduction

A major cause of the failure of human pregnancies is first trimester miscarriage [1,2,3]. Alongside endocrine and anatomical abnormalities, acquired thrombophilia or environmental agents which induce spontaneous abortion, chromosomes are an important factor which can determine the fate of embryos [1,4,5,6,7,8]. Human embryo chromosome abnormalities induce over 50% of miscarriages in the first trimester [9,10,11]. The primary chromosome error is autosome trisomy. In addition, monosomy and triploids are also common in miscarriage samples [12,13]. It is essential to identify the cause of a miscarriage in order to guide the preparation for further pregnancies or medical intervention for the couples who suffered the miscarriage.

The technologies for miscarriage analysis have become more and more accurate in the last few decades. The most classic and oldest test utilized for miscarriage analysis is cytogenetic karyotyping [14,15], followed by fluorescent in situ hybridization (FISH), array comparative genomic hybridization (aCGH) and single nucleotide polymorphism (SNP) arrays, which provide higher resolution results for miscarriage analysis [16,17,18,19,20]. More recently, quantitative fluorescent polymerase chain reaction (Q-PCR) and commercial multiple primer ligation amplification (MPLA) kits have provided faster and easier methods of investigating chromosomal errors in miscarriage samples [21,22,23]. Moreover, a novel next-generation sequencing (NGS) process known as low-pass whole genome sequencing has supplied a novel way to detect chromosomal errors [24,25]. Copy number variation sequencing (CNV-Seq), which, based on low-pass whole genome sequencing, showed higher resolution and accuracy in finding chromosomal abnormalities.

However, none of the above methods were able to distinguish triploidy, uniparental diploidy and maternal cell contamination in miscarriage samples. Short tandem repeat (STR) is a core sequence of 2–6 bases. STR loci were first used as an important genetic marker in human paternity testing in the early 1990s [24,25]. To address the aforementioned issues, we used low-pass CNV-seq combined with STR panels to detect miscarriage samples for the first time. Compared with traditional karyotyping, this method not only increased the detection rate of chromosomal abnormalities in the miscarriage samples of RSA couples but was also able to trace the parental origin of abnormal chromosomes. Compared with traditional karyotype detection methods, it also has the advantages of lower cost and a shorter detection cycle. The clinical significance of this method is that it can quickly determine whether RSA is caused by aneuploidy, polyploidy or uniparental diploidy, as well as the parental source of abnormal chromosomes, and is able to provide more sufficient clinical diagnostic information for determining whether RSA patients will require preimplantation genetic testing for aneuploidies (PGT-A) in preparation for their next pregnancy.

## 2. Materials and Methods

### 2.1. Participants

Miscarriage samples were obtained from 500 couples who underwent relevant treatment and surgery at the Shanghai JIAI Genetics and IVF Institute and the Obstetrics and Gynecology Hospital of Fudan University. Samples from both parents were collected in most cases, although in eight cases, samples from only one parent were acquired. Aborted villi were collected from miscarriage samples and peripheral blood or saliva samples were collected from the parents. Samples were collected with written informed consent from all patients under Institutional Review Board-approved protocols.

### 2.2. DNA Extraction and CNV-Seq

Tissue DNA was extracted using a QIAamp DNA kit (QIAGEN); blood DNA was extracted using a TIANamp Blood DNA Kit (TIANGEN); and saliva DNA was extracted using a commercial kit. All procedures were carried out according to the kits’ instructions.

Low-pass genome sequencing for copy number variation sequencing (CNV-seq) was carried out using 500 ng genomic DNA for the origin template. Before libraries were constructed, DNA was fragmented to an average size of about 250 bp by Smearase^®^ Mix from Hieff NGS^®^ Fast-Pace DNA Fragmentation and Ligation kit (YEASEN), according to the procedures provided by the manufacturers. Then, the fragments were ligated to 6-bp barcode adaptors using the same kit and the modified fragments were purified. After purification, PCR amplification was performed on the modified DNA. Thereafter, the amplified libraries were purified and the qualified libraries were sequenced using the Illumina higseq4000 via the PE150 sequencing strategy. The sequencing data were analyzed using bioinformatics procedures. In brief, the qualified data were compared with the reference sequences of GRCh37/hg19 and rearranged accordingly. Then, the raw sequencing data were cleaned, e.g., by removing duplications. The CNV reference was built with normal samples through CNV kit software. This method uses both on-target reads and off-target reads captured unspecifically in order to uniformly infer copy numbers in the genome. This combination enables both exon-level resolution in the target region and sufficient resolution in large intronic and intergenic regions in order to identify copy number changes. The preparation rate of this assay is generally consistent with that of conventional karyotyping methods, such as aCGH and FISH [25,26,27,28]. The data from tested samples were compared with control data, and then the CNV result was found. The resolution of our CNV-seq is 100 Kb.

### 2.3. NGS-Based STR Test

We designed an STR panel (WHS STRseq v1.0) according to 386 STR loci distributed across 22 autosomes and 2 sex chromosomes (X and Y) chromosomes (Figure 1). Before sequencing, we amplified the target regions using a WHS STRseq v1.0 kit. The following experiments were then performed: First, quality control was carried out. According to the principles of forward and reverse primer sequencing matching, the reads that did not match the forward and reverse primers (mostly primers or adapter dimers) were re-moved and the STR site names were marked. Cutadapt (version 3.2) was used to remove the splice sequence. Ctbest (Version: 1.0.0) was used to remove repetitive sequences and count the number of sequences at different sites. Secondly, the genotype was identified. The fastq sequence was converted into a fasta format file recognized by MISA (version 1.0) as the input, and STR genotypes were generated and analyzed. Lastly, the Mendelian inheritance law was used for analysis and judgment. The pedigree analysis of STR results was performed according to the Mendelian inheritance law, the possibility of triploidy was analyzed and certain special circumstances (such as family information errors, contamination, etc.) were excluded. ColoredChromosomes (Version 1.1.1) was used to create the distribution map of STR on chromosomes. For samples with CNV results that were positive for trisomy or tetrasomy, the origin of trisomy or tetrasomy (paternal or maternal) was analyzed (a product designed for multiplex PCR).

### 2.4. Chromosome G-Banding Analysis

In this study, the chorionic villi of the products were isolated and cultured in vitro. The samples were placed in 5 mL of medium (pH7.0) and cultured in CO_2_ at 37 °C for 24–48 h. Then, the cultures were prepared in two Leighton tubes (one containing half Chang DR and the other containing Amniomax R (2.5 mL, pH7.0)) and incubated in CO_2_ at 37 °C. When sufficient tissue growth was present, 60 μL of colchicine was added per 5 mL of culture for 45 min; finally, the growth medium was removed and 1% sodium citrate was added at room temperature for 10 min. The above steps were repeated twice before plates were prepared and stained with G-bands.

## 3. Results

### 3.1. Construction of an NGS-Based STR Panel

CNV-seq cannot identify uniparental disomy. Therefore, the STR panel was first introduced to detect samples with diploid CNV results. In addition, the origin of abnormal chromosomes cannot be traced through traditional analysis methods of miscarriage samples. However, the STR panel designed in this study is able to accurately trace the origins of abnormal chromosomes, such as trisomy, triploidy and monosomy. In order to improve the accuracy of chromosome tracing, a total of 386 STR loci were developed on twenty-two autosomes and two sex chromosomes (X and Y chromosomes). In contrast to conventional paternity testing, which only detects 1–2 STR loci per chromosome, 6–20 STR loci were developed on each chromosome in this study. Chromosome length and sequence structure are the constraints for finding sufficient STR loci for certain chromosomes, such as chromosomes 19 and 21 (Figure 1). Introducing a large number of STR loci to each chromosome is helpful for clinically determining whether the erroneous chromosome that leads to aneuploid abortion originates from the father or mother, which can narrow the scope of genomic examination and identify the gene mutations of the parents more precisely.

The detection of a large number of STR loci (about 6–20 STR loci per chromosome) for haplotype analysis.

### 3.2. CNV-Seq Combined with the STR Panel Analysis of the Results of Double Trisomy

A typical human has two different haplotypes, one from parental origin and the other from maternal origin. In our study, we defined the different haplotypes in maternal samples, such as M1/M2, and in paternal samples, such as P1/P2. Therefore, the haplotype of a typical fetus should be MP (M representing M1 or M2 and P representing P1/P2). In trisomy samples, the extra chromosome has either paternal origin or maternal origin. In STR results, we defined the paternal allele as “P” and the maternal allele as “M”. Moreover, the numbers “1” and “2” were used to identify two different haplotypes. Taking trisomy results as an example, we are able to identify the origin of extra chromosome through STR detection. According to the STR results, the haplotypes of trisomy can be identified as four different types, namely P1P2M1, P1P1M1, M1M2P1 or M1M1P1. When 100% of the STR results (that is, all STR loci that can be used for practical detection) demonstrated that the haplotype of the two chromosomes came from the different alleles of one parent, this result was reported as P1P2M1 or M1M2P1. When the STR results showed that the haplotype of the two chromosomes came from the same allele of one parent, this result was reported as P1P1M1 or M1M1P1. On the other hand, if the distribution of tested STR could not distinguish the two paternal/maternal alleles, the report would be PPM, where the distribution of STR could not be judged as P1P2M1 or P1P1M1 with 100% certainty, or MMP, which indicates that the extra chromosome was of paternal origin but the origin of haplotype with the higher resolution could not be identified; this is also the case for maternal origin tags M1M2P1, M1M1P1 and MMP. When designing STR loci, many sites will be constructed but not every site can be used in all samples, and some sites may be unavailable due to sample or sequencing quality. For instance, the sample No. 4 showed double trisomy (chromosome 13 trisomy and chromosome 21 trisomy) from CNV analysis (Figure 2A). We developed eleven STR loci on chromosome 13 and nine STR loci on chromosome 21, respectively. Subject to actual testing conditions, only six and seven of these were used for testing (Figure 2B–D). The STR results showed that the fetal chromosome 13 was P1P2M1, because only the STR loci of this type were 100% identical to the STR loci on the respective alleles of the parents. The STR results demonstrated that the chromosome type of chromosome 21 in the fetus was MMP, because the corresponding STR loci of M1M1P1 and M1M1P2 were only 83% and 50% of those of the parents, respectively (Appendix A). This indicates that the extra chromosome 21 was of maternal origin, but the results cannot separate the two maternal alleles. These methods help to increase the number of chromosomal abnormalities detected in miscarriage samples, which can help to determine the forms of genetic testing needed before the next pregnancy.

### 3.3. CNV-Seq Combined with the STR Panel Analysis of the Results of UDP

The detection system in this study is able to accurately distinguish the products of uniparental diploid miscarriage samples, which make up for the disadvantage of CNV-seq not being able to distinguish UDP. For example, chromosome G-banding karyotype analysis showed that the sample was an XXX uniparental diploid. CNV-seq was only able to detect diploid karyotype in this sample (Figure 3A). Through the STR analysis of this sample, of the 217 STR loci, 214 (99%) on chromosome 1 and chromosome 22 demonstrated XXX hydatidiform mole samples and the X chromosome had two copies, both of which came from the father with the same haplotype, P1, whose haplotype composition was P1P1. It was speculated that the XXX grapevine FACE sample was paternally uniparental diploid and its chromosome haplotype combination was P1P1 (Figure 3B). We can therefore conclude that the sample is XXX uniparental diploid and all chromosomes are derived from the father. This method greatly expands the karyotype detection of embryo samples from spontaneous abortions. UDP cannot be identified in the traditional single detection methods. Our method can help to clinically and quickly determine whether the euploid abortion samples are UDP, which provides an extremely important point of reference for medical guidance during future pregnancies.

### 3.4. CNV-Seq Combined with the STR Panel Analysis of the Results of Triploidy

The detection system in this study is able to accurately identify triploidy in miscarriage samples while aCGH cannot. For example, G-banding karyotype analysis showed that the sample was a 69, XXY triploid (Figure 4A,B). The detection of holotriploidy by FISH is expensive, has a long cycle and is a cumbersome process. The accuracy of CNV-seq in detecting full triploids is relatively low, however, since the introduction of STR detection, the type of triploid can be more accurately determined. CNV-seq also detected the same result. About 30% of the STR loci were consistent with the parental haplotype and the paternal haplotype, and about 60% of the STR loci were consistent with the parental haplotype and the maternal haplotype (Figure 4C). In summary, the sample in question displayed MMP triploidy and the supernumerary chromosome was derived from the mother (Figure 4B). Spontaneous RSA abortions caused by fetal triploidy accounted for about 8% of miscarriages, second only to RSA caused by aneuploidy [12,13]. Until now, there has been no rapid and accurate technique to detect triploidy in clinical practice; however, our new method will solve this problem perfectly. Our method can quickly detect the origin of abnormal chromosome sets and provide a more comprehensive guarantee for karyotype detection before embryo implantation.

### 3.5. Results of CNV-Seq Combined with the STR Panel of 500 Unexplained RSA Miscarriage Samples

We investigated CNV in miscarriage samples. It was the chromosomal-level abnormalities that were found to be the most prevalent genetic errors in miscarriages from our collected samples. The total number of samples included in this study was 500, since there was sometimes more than one sample per couple (such as samples from twins or samples from more than one spontaneous abortion from a couple). After CNV identification, 216 samples of normal chromosomes were included and 282 samples of faulty chromosomes and 2 samples with maternal contamination were filtered out. Among the erroneous chromosome samples, chromosomal aneuploidy was the most prevalent error, found in 169 trisomy samples (33.4% in total samples and 59.9% in the erroneous chromosome group), 40 triploid samples (8.0% in total samples and 14.2% in the erroneous chromosome group), 32 monosomy samples (6.4% in total samples and 11.3% in the erroneous chromosome group), 16 UPD samples (3.2% in total samples and 5.7% in the erroneous chromosome group) and 6 double trisomy samples (1.2% in total samples and 2.1% in the erroneous chromosome group), respectively. One sample were carried X monosomy and trisomy 13. Samples with Chromosome segmental microduplication or microdeletion were detected in the other 12 samples (Table 1).

Together with low-pass CNV analysis, the additional STR test showed that 94.7% of extra chromosomes in trisomy had a maternal origin. The ratio of extra paternal chromosomes was only 5.3%. The level of trisomy of maternal origin was almost 18 times more common than trisomy of paternal origin (Appendix A). Triploidy is the second most frequent aneuploidy in this study. In 40 total triploid samples, STR results indicated that 31 (77.5%) of the triploid samples’ extra chromosomes were of maternal origin and 9 (22.5%) of the extra triploid chromosomes were of paternal origin. The number of triploids of maternal origin was about 3.5 times more common than those paternal origin. It is hard to separate different parental haplotypes in triploids, so the tags for the origins of triploid are MMP or PPM (Appendix A). The third most prevalent chromosome error in this study is monosomy. The most common monosomy is 45XO monosomy, while 29 samples with X-monosomy and 3 samples with monosomy 21 were found. Among these monosomy samples, 16 X-monosomy chromosomes with a single X chromosome were of maternal origin and 13 X-monosomy chromosomes with a single X-chromosome were of paternal origin. Three samples carried only one chromosome 21 of paternal origin. The ratio of the remaining chromosomes of maternal origin and paternal origin were 50% and 50%, i.e., a 1 to 1 ratio for maternal origin to paternal origin (Appendix A). Among the 16 UPD errors found in this study, three samples with chromosomes of maternal origin only were found, while the other 13 were found to have paternal origin chromosomes only. Aside from nine whole genome UPD samples, two UPDs on chromosome 18 (both of the chromosome 18s were of maternal origin) and four UPDs on chromosome 11 (both of the chromosomes 11s were of paternal origin) were found. The ratio of UPD samples with all paternal origin chromosomes was 81%, which is four times higher than maternal chromosomes’ UPD (Appendix A). Of all the miscarriage samples, we found six that carried double trisomy errors. Among the samples with double trisomy, two (No. 3 and No. 4) had two extra chromosomes: one was of maternal origin, while the other was of paternal origin. In the other four samples with double trisomy, their abnormalities originated from only one parent. The sample carrying both monosomy and trisomy displayed a single X from maternal origin and an extra chromosome 13 of maternal origin. The STR result demonstrated that the haplotype of the extra chromosome 13 was M1M2P1 (Appendix A). Compared with conventional karyotyping methods, our new method increased the detection rate of abnormal flow karyotypes to 56.4%. Moreover, it has the ability to detect triploidy and UDP and identify maternal contamination, which cannot be achieved through traditional FISH and aCGH detection methods. This system provides a new option for the clinical identification of multiple abnormal karyotypes and, at the same time, is extremely efficient and cheap.

## 4. Discussion

In this study, we combined NGS-based STR detection with the CNV-seq in the analysis of miscarriage samples for the first time. A novel system combining STR sequencing and CNV-seq not only improved the detection rate of chromosomal errors to 56.4% but also revealed the parental origin of these errors. According to the results of STR loci, we identified the samples’ haplotypes in the detected abnormal chromosomes. After comparing the haplotypes of abnormal chromosomes from the miscarriage samples and the parents, we were able to uncover the origins of the abnormal chromosomes. Our STR test was also able to distinguish UPD samples in the group with normal CNV results. In addition, we were also able to discover polyploid samples and differentiate maternal contamination through STR testing. Our new system was also able to distinguish certain special triploids, such as “69, XXX” and “69, XXY”, which cannot be distinguished by aCGH, FISH or CNV tests alone.

Homologous recombination is indispensable in the process of human gamete formation [29,30,31]. This homologous recombination determined that the alleles in the gamete cells were not exactly the same as their hosts’ alleles [15,29,32]. In the trisomy group in this study, STR tests observed M1M1P1 haplotypes, which means that the haplotype of the extra chromosomes was the same as the other maternal chromosomes in the results. However, it cannot be claimed that this type of trisomy (M1M1P1/P1P1M1) error occurs in the second meiosis of the oocytes, since the STR in the pericentromeric region were not included in this panel. For the M1M2P1 or P1P2M1 trisomy samples, we also cannot speculate at which stage of meiosis the errors occurred. It is hard to distinguish whether the two chromosomes (such as MM and PP) belong to M1/M2 or P1/P2 in MMP or PPM types, which may be explained by the fact that homologous recombination induced the differences in STR. However, we are at least able to uncover the origins of extra chromosomes, which can help distinguish whether an error originates from an oocyte or sperm.

Beside trisomy, we are also able to differentiate between the origins of abnormal chromosomes in triploidy, monosomy, UPD and double trisomy. In the triploidy results, the origin of 77% of the extra chromosomal copies was maternal. The reasons for triploid formation are complicated—it may result from meiosis errors in the gametes or mitosis errors in early embryogenesis [33,34,35]. The results regarding the maternal origins of triploidy may indicate meiosis errors in oocyte formation. However, the mitosis errors during the early fertilization cannot be excluded. In terms of UPD, STR results revealed regions and chromosomes with UPD. We found nine whole genome UPD samples in this study, and STR results demonstrated that eight of these displayed paternal whole genome UPD (Appendix A). Paternal whole genome UPD samples displayed complete hydatidiform mole phenotypes, which may be caused by oocyte errors, such as empty oocyte fertilization, or other, unknown, factors. Maternal whole genome UPD displayed teratoma phenotypes, which may result from oocyte errors [36,37]. Interestingly, based on the STR analysis, we found six double trisomy samples in this study, and two extra chromosomes of two samples were of maternal and paternal origin, respectively (Appendix A). Most double trisomy cases result in miscarriage and the few live birth neonates suffered a multitude of abnormalities, such as intrauterine growth retardation, facial abnormalities, visceral abnormalities, etc. [38,39,40,41,42].

Compared with the traditional karyotype analysis technology, our method improved the detection rate of abnormal karyotypes in abortion products. Compared with aCGH, this method is able to detect triploidy and UDP. In contrast to FISH, this method is able to detect maternal cell contamination in the miscarriage samples. The system is suitable for a wide range of samples, including blood, tissue, saliva, cells and paraffin. In the future, we will improve the detection method and attempt to apply it to assisted reproductive technology in order to help doctors quickly identify the karyotypes of embryos from spontaneous abortions and determine whether to perform PGT-A testing in preparation for subsequent pregnancies.

## Figures and Tables

**Figure 1 jcm-12-01809-f001:**
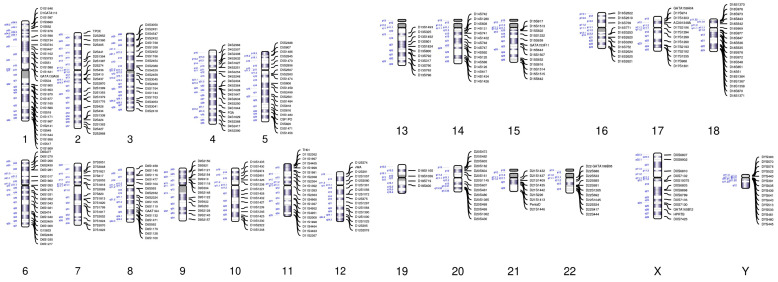
STR loci developed on each human chromosome.

**Figure 2 jcm-12-01809-f002:**
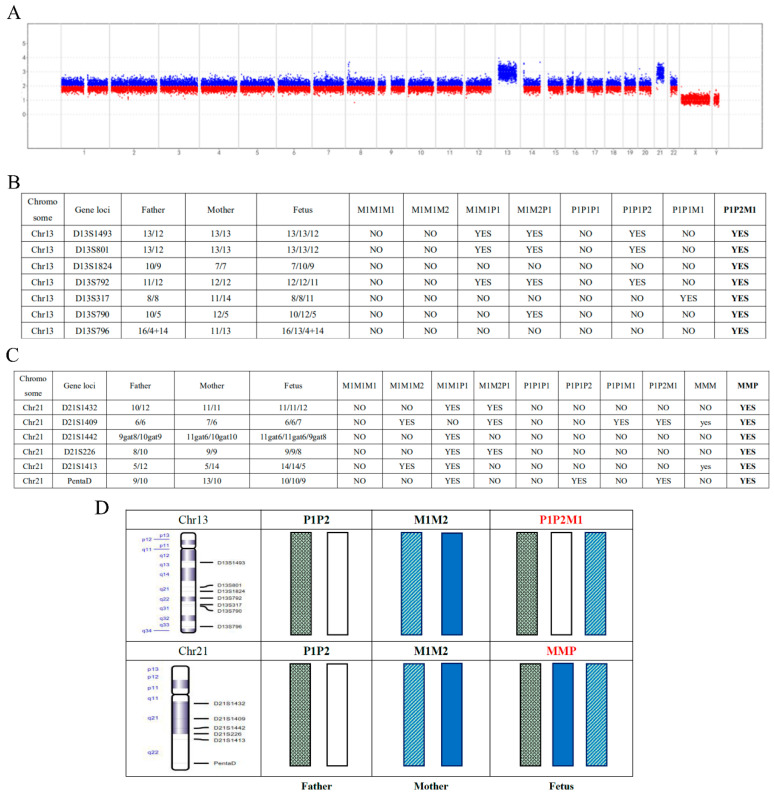
No. 4 sample displays double trisomy (chromosome 13 trisomy and chromosome 21 trisomy) from CNV and STR analysis. (**A**) The No. 4 sample CNV-Seq results shows that chromosomes 13 and 21 have double trisomy. (**B**,**C**) Fetuses are compared with the STR markers of the father at the loci on chromosomes 13 and 21. For instance, “13/12” means the number of duplicate segments of D13S1493 on both alleles. The black bold means that the STR loci on this chromosome correspond in percentage to this type. The label “13/12” represents a pair of alleles with 13 and 12 repeat occurrences of the same sequence in the developed STR loci, respectively. The “9gat8” label represents a gene with a pair of alleles. In the developed STR loci, the number of repeats of the same sequence before and after gat bases is nine and eight, respectively. (**D**) The physical map of gene loci containing the STR loci on chromosomes 13 and 21 (M: mother haploid; P: father haploid).

**Figure 3 jcm-12-01809-f003:**
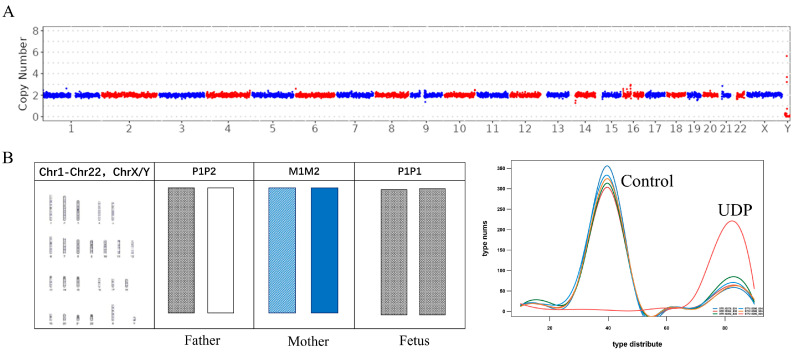
Results of CNV-seq and STR panel detection of UDP miscarriage samples. (**A**) Plot of CNV-seq detection results of UDP. (**B**) Chromosomal G-banding and STR detection results of miscarriage sample UDP. The control curve represents the proportion of chromosomal STR loci tested that are identical to the constructed chromosomal haplotype STR loci of approximately 50%. The UDP curve represents approximately 100% of the detected chromosomal STR loci identical to the constructed chromosomal haplotype STR loci.

**Figure 4 jcm-12-01809-f004:**
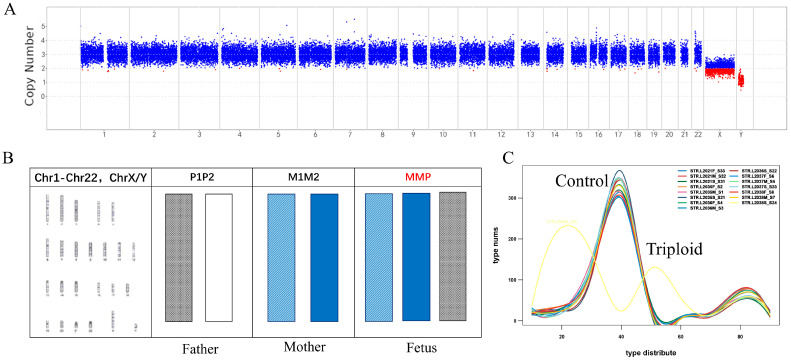
Results of CNV-seq and STR panel detection of triploid miscarriage samples. (**A**) Plot of CNV-seq detection results of the 69, XXY triploid. (**B**) Chromosomal G-banding and STR detection results of miscarriage triploidy. (**C**) The control curve represents the proportion of chromosomal STR loci tested that are identical to the constructed chromosomal haplotype STR loci of approximately 50%. The triploidy curve represents the concordance of approximately 30% of the detected chromosomal STR loci with the constructed paternal chromosomal haplotype STR loci and approximately 60% with the constructed maternal chromosomal haplotype STR loci.

**Table 1 jcm-12-01809-t001:** Summary of CNV test results.

	Count	Ratio in Total Samples	Ratio in Abnormal Samples
Trisomy	169	33.4%	59.9%
Double trisomy	6	1.2%	2.1%
Triploid	40	8.0%	14.2%
Monosomy	32	6.4%	11.3%
UPD	16	3.2%	5.7%
Chromosome segmental microduplication or microdeletion	18	3.6%	6.4%
Monosomy + trisomy	1	0.2%	0.4%
Maternal contamination	2	0.4%	/
Normal	216	43.2%	/
Total	500		
Abnormal total	282	56.4%	

## Data Availability

The obtained data would be disclosed upon requests. However, information for the patients results has been removed for confidential reasons.

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
