# Peer review of "A Novel System for the Detection of Spontaneous Abortion-Causing Aneuploidy and Its Erroneous Chromosome Origins through the Combination of Low-Pass Copy Number Variation Sequencing and NGS-Based STR Tests"

_jcm, 2023, doi:10.3390/jcm12051809_

Round 1

Reviewer 1 Report

Thank you for submitting this manuscript of an original study pertaining to "A Novel System for the detection of aneuploidy causing abortion and its error-chromosome origin through the Combination of Low Pass Copy Number variation Sequence and NGS-Based 3 STR test".

I recommend accepting with major changes.

We note the following issues to be resolved:

A. Major changes:

- English language needs to be revised extensively.

- The third paragraph of the introduction includes results and conclusion information, please remove. Please add clinical significance of the study in the introduction.

- Important to note if the patients included in the study were pregnant from natural conception or ART. If ART, was there any PGT-A performed on the embryo prior?

- Please elaborate on the clinical relevance of the findings in the conclusion.

- Need to add paragraph about statistical analysis

- Need to add conclusion

B. Minor Changes:

- STR: needs to be defined the first time it appears in text 

- Although it is not the purpose of this study, comparison of your results to the results of other types of testing would have been interested in the conclusion.

- Line 259-262 should be in the results part not in the discussion

Author Response

  1. Major changes:

- English language needs to be revised extensively.

We have used your recommended MDPI paid retouching service and obtained the certificate

- The third paragraph of the introduction includes results and conclusion information, please remove. Please add clinical significance of the study in the introduction.

We have revised the article in full accordance with your suggestion.

However, none of the above methods were able to distinguish triploidy, uniparental diploidy and maternal cell contamination in miscarriage samples. Short tandem repeat (STR) is a core sequence of 2–6 bases. STR loci were first used as an important genetic marker in human paternity testing in the early 1990s [24,25]. To address the aforementioned issues, we used low-pass CNV-seq combined with STR panels to detect miscarriage samples for the first time. Compared with traditional karyotyping, this method not only increased the detection rate of chromosomal abnormalities in the miscarriage samples of RSA couples but was also able to trace the parental origin of abnormal chromosomes. Compared with traditional karyotype detection methods, it also has the advantages of lower cost and a shorter detection cycle. The clinical significance of this method is that it can quickly determine whether RSA is caused by aneuploidy, polyploidy or uniparental diploidy, as well as the parental source of abnormal chromosomes, and is able to provide more sufficient clinical diagnostic information for determining whether RSA patients will require preimplantation genetic testing for aneuploidies (PGT-A) in preparation for their next pregnancy.

- Important to note if the patients included in the study were pregnant from natural conception or ART. If ART, was there any PGT-A performed on the embryo prior?

The majority of patients conceived naturally. None of the patients had previously undergone PGT-A testing. Because of the lengthy duration of PGT-A testing, it is not necessary to perform PGT-A testing until RSA due to aneuploidy is confirmed.

- Please elaborate on the clinical relevance of the findings in the conclusion.

We have revised the article in full accordance with your suggestion.

Compared with the traditional karyotype analysis technology, our method improved the detection rate of abnormal karyotypes in abortion products. Compared with aCGH, this method is able to detect triploidy and UDP. In contrast to FISH, this method is able to detect maternal cell contamination in the miscarriage samples. The system is suitable for a wide range of samples, including blood, tissue, saliva, cells and paraffin. In the future, we will improve the detection method and attempt to apply it to assisted reproductive technology in order to help doctors quickly identify the karyotypes of embryos from spontaneous abortions and determine whether to perform PGT-A testing in preparation for subsequent pregnancies.

- Need to add paragraph about statistical analysis

We have revised the article in full accordance with your suggestion.

We investigated CNV in miscarriage samples. It was the chromosomal-level abnormalities that were found to be the most prevalent genetic errors in miscarriages from our collected samples. The total number of samples included in this study was 500, since there was sometimes more than one sample per couple (such as samples from twins or samples from more than one spontaneous abortion from a couple). After CNV identification, 216 samples of normal chromosomes were included and 282 samples of faulty chromosomes and 2 samples with maternal contamination were filtered out. Among the erroneous chromosome samples, chromosomal aneuploidy was the most prevalent error, found in 169 trisomy samples (33.4% in total samples and 59.9% in the erroneous chromosome group), 40 triploid samples (8.0% in total samples and 14.2% in the erroneous chromosome group), 32 monosomy samples (6.4% in total samples and 11.3% in the erroneous chromosome group), 16 UPD samples (3.2% in total samples and 5.7% in the erroneous chromosome group) and 6 double trisomy samples (1.2% in total samples and 2.1% in the erroneous chromosome group), respectively. One sample were carried X monosomy and trisomy 13. Samples with chromosomal fragment microduplication or microdeletion were detected in the other 12 samples (Table1)

- Need to add conclusion

We have revised the article in full accordance with your suggestion.

3.4. CNV-seq combined with the STR panel analysis of the results of triploidy

The detection system in this study is able to accurately identify triploidy in miscarriage samples while aCGH cannot. For example, G-banding karyotype analysis showed that the sample was a 69, XXY triploid (Figure 4A,B). The detection of holotriploidy by FISH is expensive, has a long cycle and is a cumbersome process. The accuracy of CNV-seq in detecting full triploids is relatively low, however, since the introduction of STR detection, the type of triploid can be more accurately determined. CNV-seq also detected the same result. About 30% of the STR loci were consistent with the parental haplotype and the paternal haplotype, and about 60% of the STR loci were consistent with the parental haplotype and the maternal haplotype (Figure 4C). In summary, the sample in question displayed MMP triploidy and the supernumerary chromosome was derived from the mother (Figure 4B). Spontaneous RSA abortions caused by fetal triploidy accounted for about 8% of miscarriages, second only to RSA caused by aneuploidy [12,13]. Until now, there has been no rapid and accurate technique to detect triploidy in clinical practice; however, our new method will solve this problem perfectly. Our method can quickly detect the origin of abnormal chromosome sets and provide a more comprehensive guarantee for karyotype detection before embryo implantation.

Figure 4. Results of CNV-seq and STR panel detection of triploid miscarriage samples. (A) Plot of CNV-seq detection results of the 69, XXY triploid. (B) Chromosomal G-banding and STR detection results of miscarriage triploidy. The control curve represents the proportion of chromosomal STR loci tested that are identical to the constructed chromosomal haplotype STR loci of approximately 50%. The triploidy curve represents the concordance of approximately 30% of the detected chromosomal STR loci with the constructed paternal chromosomal haplotype STR loci and approximately 60% with the constructed maternal chromosomal haplotype STR loci.

  1. Minor Changes:

- STR: needs to be defined the first time it appears in text

We have revised the article in full accordance with your suggestion.

However, none of the above methods were able to distinguish triploidy, uniparental diploidy and maternal cell contamination in miscarriage samples. Short tandem repeat (STR) is a core sequence of 2–6 bases. STR loci were first used as an important genetic marker in human paternity testing in the early 1990s [24,25].

- Although it is not the purpose of this study, comparison of your results to the results of other types of testing would have been interested in the conclusion.

We have revised the article in full accordance with your suggestion.

Compared with the traditional karyotype analysis technology, our method improved the detection rate of abnormal karyotypes in abortion products. Compared with aCGH, this method is able to detect triploidy and UDP. In contrast to FISH, this method is able to detect maternal cell contamination in the miscarriage samples.

- Line 259-262 should be in the results part not in the discussion

We have revised the article in full accordance

Reviewer 2 Report

Thank you for the opportunity to review this manuscript which examines using a combination of low pass copy number variation sequence and NGS-based STR testing for detecting aneuploidy causing early pregnancy loss.  There were some grammar/language errors throughout that could benefit by someone with a better understanding of the English language

Introduction:

-what do the authors mean by “Human embryos, in early stage, always suffer from chromosomal errors”   This is not true

 -authors quote 50% of miscarriages in the first trimester are because of chromosomal abnormalitiesàthis is low, with more recent studies stating a higher percentage (70-80%)

-It is indispensable to identify the cause of miscarriage to guide further preparation of next pregnancy or medical intervention for the couples who suffered miscarriage” no “-“ is needed for preparation;  most societies do not advocate for fetal karyotype for a first time pregnancy lossàplease qualify your statement

Material and methods:

“Aborted villi were collected from 84 miscarriage samples”:  what do the authors mean by “aborted villi”?  were these samples collected after a D+C procedure (dilation and curettage?)

Results:

3.3:  the authors give an example of how a molar pregnancy from paternal origin was detected;  the authors can state this example, but should not use “Conclusion:”  (they can state it was concluded that…..)

-were there any samples that had both maternal and paternal origins of chromosomal error?

Editor note:

There are many words that are hyphenated (-) that should not be

Author Response

Reviewer 2

Introduction:

-what do the authors mean by “Human embryos, in early stage, always suffer from chromosomal errors”   This is not true

We have revised the article in full accordance with your suggestion.

A major cause of the failure of human pregnancies is first trimester miscarriage [1-3]. Alongside endocrine and anatomical abnormalities, acquired thrombophilia or environmental agents which induce spontaneous abortion, chromosomes are an important factor which can determine the fate of embryos [1],[4-8]. Human embryo chromosome abnormalities induce over 50% of miscarriages in the first trimester [9-11].

 -authors quote 50% of miscarriages in the first trimester are because of chromosomal abnormalitiesàthis is low, with more recent studies stating a higher percentage (70-80%)

We have revised the article in full accordance with your suggestion. Human embryo chromosome abnormalities induce over 50% of miscarriages in the first trimester [9-11].

-It is indispensable to identify the cause of miscarriage to guide further preparation of next pregnancy or medical intervention for the couples who suffered miscarriage” no “-“ is needed for preparation;  most societies do not advocate for fetal karyotype for a first time pregnancy lossàplease qualify your statement

We do not advocate karyotyping of embryos on initial IVF but rather recommend PGT-A for embryos from patients with a history of RSA due to embryo aneuploidy.

Material and methods:

“Aborted villi were collected from 84 miscarriage samples”:  what do the authors mean by “aborted villi”?  were these samples collected after a D+C procedure (dilation and curettage?)

 Most of the aborted villi were obtained surgically after clinical confirmation of fetal death.

Results:

3.3:  the authors give an example of how a molar pregnancy from paternal origin was detected;  the authors can state this example, but should not use “Conclusion:”  (they can state it was concluded that…..)

 We have revised the article in full accordance with your suggestion.

We can therefore conclude that the sample is XXX uniparental diploid and all chromosomes are derived from the father. This method greatly expands the karyotype detection of embryo samples from spontaneous abortions. UDP cannot be identified in the traditional single detection methods. Our method can help to clinically and quickly determine whether the euploid abortion samples are UDP, which provides an extremely important point of reference for medical guidance during future pregnancies.

-were there any samples that had both maternal and paternal origins of chromosomal error?

 Yes,in result 3.2, the sample No.4 showed double trisomy (chromosome 13 trisomy and chromosome 21 trisomy), the extra chromosome 13 came from paternal origin while the extra chromosome 21 came from maternal origin.

Editor note:

There are many words that are hyphenated (-) that should not be

We have revised the article in full accordance with your suggestion.
